# Grasping extreme aerodynamics on a low-dimensional manifold

Kai Fukami ⬤ [1] & Kunihiko Taira ⬤ [1] ✉

Modern air vehicles perform a wide range of operations, including transportation, defense, surveillance, and rescue. These aircraft can fly in calm conditions but avoid operations in gusty environments, encountered in urban canyons, over mountainous terrains, and in ship wakes. With extreme weather becoming ever more frequent due to global warming, it is anticipated that aircraft, especially those that are smaller in size, will encounter sizeable atmospheric disturbances and still be expected to achieve stable flight. However, there exists virtually no theoretical fluid-dynamic foundation to describe the influence of extreme vortical gusts on wings. To compound this difficulty, there is a large parameter space for gust-wing interactions. While such interactions are seemingly complex and different for each combination of gust parameters, we show that the fundamental physics behind extreme aerodynamics is far simpler and lower-rank than traditionally expected. We reveal that the nonlinear vortical flow field over time and parameter space can be compressed to only three variables with a lift-augmented autoencoder while holding the essence of the original high-dimensional physics. Extreme aerodynamic flows can be compressed through machine learning into a low-dimensional manifold, which can enable real-time sparse reconstruction, dynamical modeling, and control of extremely unsteady gusty flows. The present findings offer support for the stable flight of next-generation small air vehicles in atmosphere conditions traditionally considered unflyable.

Since the Wright brothers accomplished the first human-powered and controlled flight in 1903, a wide variety of aircraft has been developed for transportation, defense, observation, search, and rescue missions. What makes these aircraft uniquely different from other modes of transportation is their ability to stay aloft by taking advantage of aerodynamics. To support the development of these aircraft, the field of aerodynamics has undergone tremendous growth over the past century. Despite its expansive theory, the current aerodynamics is generally based on steady (cruise) or quasi-steady flight conditions with linear analysis or its nonlinear extensions for small perturbations[1–3].

We are now at an important transition point for aerodynamics. With novel materials and enhanced powerplants/batteries becoming available over the past couple of decades, there have been tremendous efforts in developing and operating smaller size personalized air vehicles and unmanned air systems[4–8]. They can traverse unconventional terrain, including mountainous and urban environments[9–11] that were traditionally avoided by conventional aircraft. As a matter of fact, flight demonstrations of such air vehicles in calm weather have taken place in recent years[12]. These new and amazing flying vehicle concepts will likely revolutionize air-based transportation[5,13,14] and have already been realized for some cases[15,16].

However, there are major challenges in operating small-scale aircraft under these complex airspace when adverse weather generates highly turbulent environments[17–21]. Moreover, the increased occurrence of extreme weather caused by global warming limits the operations of aircraft. Flying small-scale aircraft near large natural or manmade structures in adverse weather face additional complications

[1]Department of Mechanical and Aerospace Engineering, University of California, Los Angeles, CA, USA. ✉e-mail: ktaira@seas.ucla.edu

as these vehicles need to navigate through severe turbulence comprised of gusts and vortical disturbances, as illustrated in Fig. 1. These disturbances come in a variety of forms[22] and are far more disruptive than what present-day commercial aircraft experience in inclement weather[23]. These extreme aerodynamic environments can be characterized by a manifestation of a large number of strong vortices with different strengths, sizes, and orientations generated by the surrounding structures[24]. Such a flight environment has been off-limits due to the fact that there is virtually no available theory for extreme aerodynamic problems and to avoid possible loss of aircraft.

With infinite scenarios of large and strong atmospheric disturbances hitting a flying vehicle, we cannot only focus on a single cruise condition but must also consider a whole array of cases in which wings experience extreme aerodynamic disturbances. These disturbances are characterized by a variety of parameters, including the size, strength, orientation, position, and geometry of the disturbances, necessitating massive experimental and computational campaigns if approached naively. With a single extreme aerodynamic simulation already producing a very large amount of flow field data, extensive parametric sweeps lead to an enormous collection of aerodynamic flow data and calls for significant computational and experimental resources. These extreme aerodynamic flows exhibit rich nonlinear behavior over a range of spatiotemporal scales that cannot be easily analyzed and modeled with existing theories.

One of the important parameters under such extreme aerodynamic situations is the gust ratio $G = u_{gust}/u_\infty$, which is ratio between the characteristic gust velocity $u_{gust}$ and the translational velocity of the wing $u_\infty$. For conditions of $G \gtrsim 1$, sustaining stable flight becomes challenging[18,25]. In the present work, we consider high levels of aerodynamic disturbance with $0 \leq G \leq 10$, and refer to cases of $G > 1$ as *extreme aerodynamics*. Strong gusts with $G > 1$ can be encountered in urban canyons, mountainous environments, and severe atmospheric turbulence. The goal of this study is to identify the unifying dynamics that a wing experiences from extreme gust disturbances. At the most fundamental level, the present problem requires the identification of the underlying nonlinear dynamics of the complex separated flows from an enormous amount of data in an efficient manner while gaining physical insights into extreme aerodynamics.

Although the aerodynamic influence of large-scale disturbances on lifting bodies take various forms, the underlying dynamics are generally shared. In this study, we seek these dominant dynamics embedded in complex extreme aerodynamic flows. To achieve this objective, we examine the reduction of massive fluid flow data into a low-dimensional space in which the right set of variables describe the underlying extreme aerodynamic physics. This process is enabled by incorporating physical observables and ensuring that the gained insights are interpretable and beneficial for future aircraft operations and designs. In fact, we find that extreme aerodynamic flows can be compressed by a carefully designed nonlinear machine-learning technique to only three variables for a model problem of a strong

vortex hitting a canonical airfoil. The present findings further suggest that the discovered manifold holds potential to support downstream tasks such as real-time flow estimation, dynamical modeling, flow control, and vehicle design.

## Results

### Extreme vortex–airfoil Interactions

We consider a strong vortex gust impacting an airfoil as a representative model problem for wings experiencing extreme atmospheric disturbances. The present model problem involves wake vortices shedding from a high-rise building, ships in rough seas, and mountain ridges[18]. The size of such strong vortices can be comparable to the size of the wing, exerting tremendously large lift and drag forces. In this study, we simulate two-dimensional incompressible flows with a vortex placed upstream of the wing with varied vortex size, strength, and initial position, as shown in Fig. 1. Because the airfoil wake responds differently for each combination of these disturbance settings, the resulting flow fields exhibit vastly different wake patterns and aerodynamic forces from case to case due to the nonlinear vortex dynamics, as presented in Fig. 2. As the vortex passes around an airfoil, the wing experiences massive flow separation (stall), which also causes the emergence of additional vortical structures. All of these flow structures interact nonlinearly, making the dynamics complex and difficult to predict and control.

This study considers two-dimensional extreme aerodynamic flows around a NACA 0012 airfoil at a chord-based Reynolds number $Re = u_\infty c/\nu = 100$. Here, $u_\infty$ is the free-stream velocity, $c$ is the chord length, and $\nu$ is the kinematic viscosity. The flow field is obtained with direct numerical simulation using an incompressible flow solver[26,27]. The airfoil is positioned in the free stream with its leading edge at the origin with angles of attack of $\alpha \in [20°, 60°]$, enabling us to cover cases of steady and unsteady wakes in undisturbed (baseline) cases. These wakes are disturbed with a gust vortex having an angular velocity profile[28] of $u_\theta = u_{\theta,max}(r/R) \exp[1/2 - r^2/(2R^2)]$, where the radius of the vortex is $R$.

The present disturbance vortex is characterized by the gust ratio $G \equiv u_{\theta,max}/u_\infty \in [-10,10]$ with its size relative to the wing chord $L \equiv 2R/c \in [0.5, 2]$ and is introduced upstream of the wing at $x_0/c = -2$ and $y_0/c \in [-0.5, 0.5]$. The combinations of these parameters provide a wide range of large and strong gust vortices hitting the wing at various locations. The flow field around the airfoil exhibits a rich dynamical response to the disturbance vortex characterized by a large parameter space comprised of $(\alpha, G, L, y_0/c)$. To fully resolve the dynamics over this parameter space, a substantial number of flow cases for different combinations of these parameters would be required. In general, Re is also another parameter but is fixed for this study at 100. While the gust vortices contained in actual atmospheric turbulence can be much more complex than what is considered here, the primary dynamics of large vortex core interacting with the wing is captured well at this Re at least in a two-dimensional manner. What is

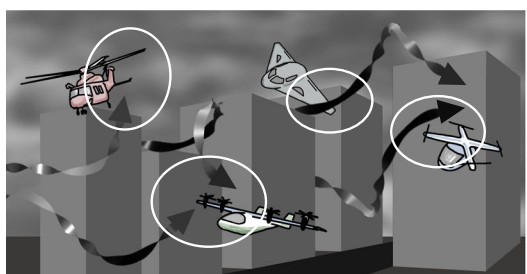

**Fig. 1 | Potential extreme aerodynamic encounters.** (Left) Illustration of possible extreme aerodynamic encounters by modern air vehicles in urban environment during adverse weather. Air vehicles operating in such an environment experience extreme level of unsteady aerodynamic forces due to strong gusts with spatial variations comparable to their vehicle size. (Right) Model problem of strong disturbance vortex impinging on an airfoil with vorticity distribution being visualized.

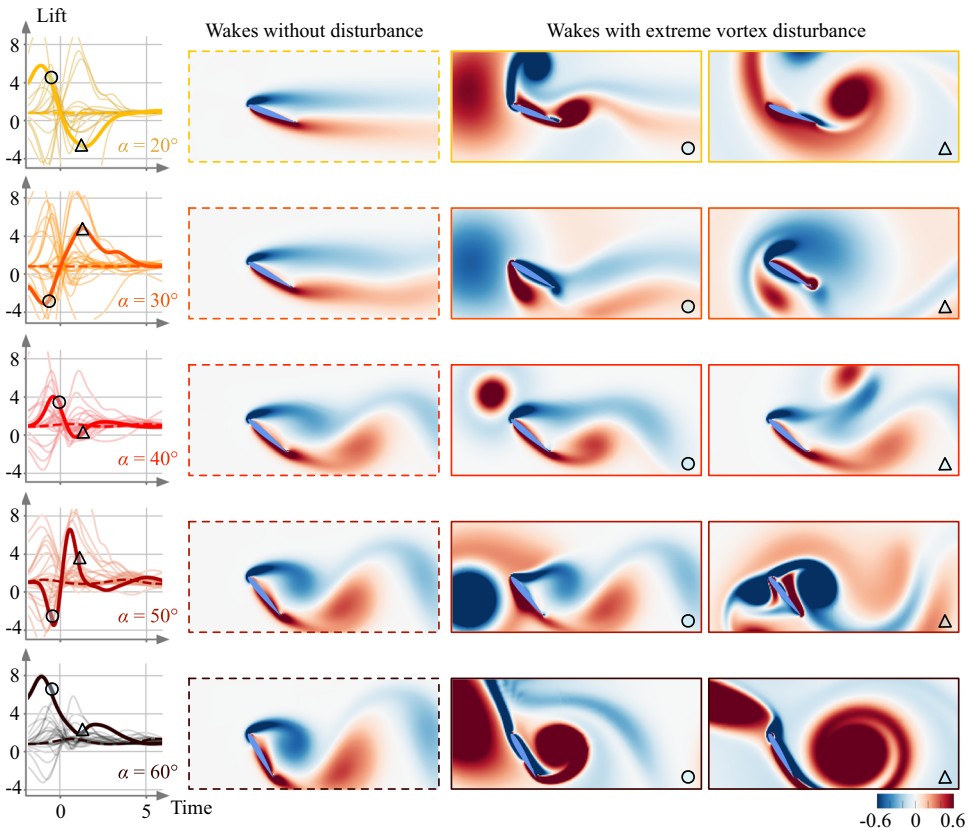

**Fig. 2 | Examples of lift responses and vortical flows.** Visualizations of the vorticity fields from times indicated by symbols ○ and △ on the lift responses (thick lines associated with the visualized cases). The light-colored lift curves correspond to all lift responses considered in the present study. The shown color for each angle of attack is shared with other figures.

particularly important is to resolve the large vorticity source from the surface under local flow acceleration[29]. These phenomena have relatively short time-scales compared to the much longer viscous scales associated with Re = 100, making the current problem setup an appropriate test bed. In this study, we set the convective time to be zero when the center of the vortex arrives at the leading edge of the wing $x_0/c = 0$. The snapshots of the vorticity field and the lift history are examined in detail with data-driven analyses in what follows.

Without any external disturbances, the wing experiences steady aerodynamic lift forces at low angles of attack ($\alpha \lesssim 20°$) and moderate unsteady aerodynamic force fluctuations at higher angles of attack ($\alpha \gtrsim 30°$). These lift values are shown by the dashed lines for the different angles of attack in Fig. 2. The unsteady lift fluctuations are exerted by the von Karman vortices shedding from the leading and trailing edges, as visualized in Fig. 2. At this Reynolds number, these cases are periodic in time and constitute limit cycles.

When the wing encounters a strong gust vortex, the flow around the wing is significantly modified. The approaching vortex strongly influences the vorticity field around the wing and triggers large-scale vortex formation. For example, let us consider the flow field for a case shown in Fig. 2 for which a strong positive vortex with $G = 3.8$ hits a wing at $\alpha = 20°$. Due to this disturbance, two large vortices are formed shortly after impact, generating massive separation due to the interaction of the gust vortex and the wing wake. These dramatic transient wake dynamics exert sharp increase in aerodynamic forces on the wing. In fact, the lift force increases 714% and drops 656% within a short duration of 1.8 convective time. Such a significant variation in the lift force makes controlling air vehicles tremendously difficult. While not shown, the moment experienced by the wing also undergoes a tremendous change. We also display a total of 100 force histories for

cases of flows disturbed by extreme levels of gust vortices in Fig. 2. In all of these cases, the airfoil wake dynamics undergo large transient changes within a short amount of time with large aerodynamic force fluctuations with similar order of magnitudes. These violent disturbances not only destabilize flight but can also damage vehicle structures, making the analysis of these flows critically important.

Because of the nonlinear nature of the dynamics, flow around the airfoil responds differently to each different gust vortex with massive flow separation and large-scale formation of additional vortices, as visualized in Fig. 2. There is no aerodynamic model or theory that can easily describe the highly nonlinear nature of the extreme gust-airfoil interactions. What is especially challenging is that there is no simple scaling that collapses the collection of lift curves or the vortical flow fields due to the strong nonlinearity. This is an enormous burden for studying extreme aerodynamic flows since each and every case needs to be examined through painstaking computational or experimental campaigns that require significant resources. We re-emphasize that even for the present problem setup, there are a good number of parameters (vortex strength $G$, size $L$, position $y_0/c$, and airfoil angle of attack $\alpha$) that necessitate a very large number of cases of extreme aerodynamic flows to map out the response dynamics. Practically speaking, such a campaign may not be possible for all gust encounters with limited computational resources. Therefore, it is desirable to capture the underlying dominant dynamics that form the basis of extreme gust response characteristics without having to rely on expensive simulations with a very large degree of freedom, which in this case is proportional to $2.88 \times 10^4$ grid (spatial) points to describe the instantaneous flow field and $1.26 \times 10^5$ temporal frames for sufficient spatiotemporal solutions for all cases. While we do have the Navier–Stokes equations as the governing partial differential

equations to fully describe the dynamics, solving them in real time for practical air vehicle operations is out of the question.

For the aforementioned reasons, it is important to extract the dominant low-dimensional dynamics from the possible collection of extreme aerodynamic data sets. The rich responses to different gust vortices appear uniquely different from one snapshot to another but possess some common and identifiable features, including the disturbance vortex, flow separation, wake vortices, secondary vortices, shear layers, vortex roll-up, vortex pinch-off, and vortex deformation. The fact that these features are indeed identifiable by the trained eyes of fluid dynamicists suggests that there is likely some underlying low-dimensional representation of the high-dimensional complex dynamics. Thus, we aim to capture the key dynamics in a space comprised of a very small number of variables that can estimate the full state of the flow field and offer insights into the nonlinear dynamics of the vortex-gust interaction. We find that a nonlinear autoencoder with physical observables illustrated in Fig. 3 achieves the present objective.

### Identification of extreme aerodynamic manifold

To find a low-dimensional space that captures the essential physics of extreme aerodynamic interactions between the gust vortex and the airfoil wake, we perform data-driven compression of the flow field. Herein, we consider cases with randomly-sampled parameters from $G \in [-4, 4]$, $L \in [0.5, 2]$, and $y_0/c \in [-0.5, 0.5]$.

First, let us consider the most commonly used linear dimensionality reduction technique, namely the principal component analysis (PCA), which is also known as the proper orthogonal decomposition (POD)[30-32]. With this method, the low-dimensional representation of a fluctuating variable is found by identifying the primary modes (or vectors) that best capture the variance about the mean. In the present study, we first analyze the vorticity fields over $(x, y)/c \in [-1.4, 4] \times [-1.2, 1.2]$, shown in Fig. 2 by applying PCA to determine the most vortically energetic components of the flow field.

Shown in Fig. 4 are the temporal variations of the first three PCA components ($\xi_1(t)$, $\xi_2(t)$, and $\xi_3(t)$) of the vorticity field data. The gray curves represent the whole collection of extreme aerodynamic data plotted in this coordinate space. Also highlighted are undisturbed baseline cases for $\alpha = 20°$ to $60°$. Here, we observe that the gray curves span a range of values over $\xi_1$, $\xi_2$, and $\xi_3$ in a seemingly incoherent manner. What is further problematic with this compression is the overlap of the baseline cases. These observations reveal that PCA struggles to compress the extreme aerodynamic data in a meaningful manner while keeping different angles of attack cases distinct. Because different flows over different angles of attack are collapsed as the same, the overlapping low-dimensional representations produced by PCA cannot distinguish important flow characteristics and yield grossly inaccurate flow reconstructions, as shown in Fig. S.1.

The challenges of reducing degrees of freedom (dimension) of extreme vorticity dynamics by PCA can be mitigated by utilizing a nonlinear compression technique. For this purpose, we utilize a nonlinear convolutional autoencoder, presented in Fig. 3 (excluding the green-shaded side network), to reduce a large number of vorticity field data to very few variables. An autoencoder is a neural network composed of an encoder and a decoder with a bottleneck in the middle[33-35]. Generally, this neural network framework is used to take an input data and replicate the same data at the output. The variables that lie in the middle are referred to as latent variables (red circles in Fig. 3), which hold the compressed information about the input data. When the nonlinear autoencoder can replicate the same input data at the output, this means that both the encoder and the decoder function effectively to nonlinearly transform the full data set to a low-dimensional latent variable $\xi$ and vice versa effectively. The present autoencoder first compresses a flow field using a convolutional neural network (CNN)[36] to capture global features of the vortical flow field. The compressed vector (extracted feature) through the CNN is then flattened at the reshape layer in Fig. 3 to pass into a multi-layer perceptron (MLP)[37] towards the latent space. A similar operation is performed for the decoder side to expand the dimension of the latent variable back to the size of the original flow field. The present autoencoder is trained with the same data sets as that used for PCA. The details of the autoencoder setup used in the present study are provided in Supplemental Material.

The nonlinear autoencoder is able to compress the flow field data and reproduce the flow field accurately, as shown in Fig. S.1. We also present the latent space comprised of only three latent variables ($\xi_1, \xi_2$, and $\xi_3$) in Fig. 4 (middle). The ability of the nonlinear autoencoder to compress the vorticity field to mere three variables is not only surprising but also reaffirms that the flow field is indeed comprised of common flow features. The compression capability offered by a nonlinear autoencoder is promising to capture violent flow physics that appears tremendously rich. Nonetheless, we should note that the full data set shown in Fig. 4 is distributed over the latent space without a meaningful collapse of the latent variables. This is sufficient if the objective is to ensure that data in latent space are distinct, thus covering as much space as necessary to ensure uniqueness of the information. However, with regard to this study, we are aiming not only to compress the extreme aerodynamic flow data but also to identify universal features among the large number of flow field data holding dynamical information.

The above autoencoder analysis was performed purely from a data-centric perspective. Instead, let us consider incorporating a physical measurement (observable) into the autoencoder to facilitate the identification of a low-dimensional subspace defined by the appropriate latent variable coordinates. Capturing the low-dimensional nature of extreme aerodynamic flows can support the flight stabilization of air vehicles in extreme levels of turbulence. For this reason, we weigh the latent space variables $\xi$ with lift force acting on the wing. This is achieved by supplementing the autoencoder with a multi-layer perceptron that outputs the lift force for each vorticity field over time, as shown by the green-shaded network in Fig. 3. In this case, training is performed to compress the vorticity field to the latent variables and to estimate the lift force accurately from the latent variables.

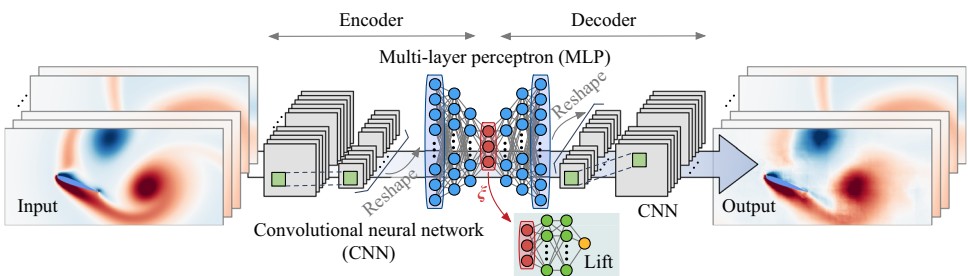

**Fig. 3 | Nonlinear autoencoder.** The vorticity field is taken as the input and output. The green shaded portion becomes active when embedding lift into the compression process.

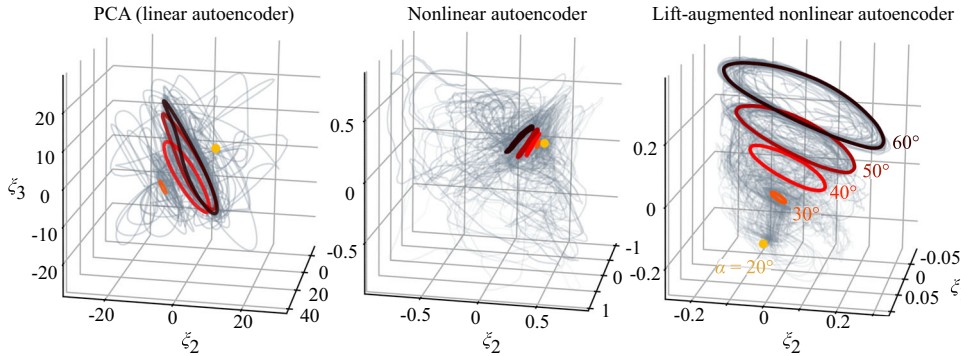

**Fig. 4 | Compression of extreme aerodynamic flow data using (left) PCA (linear autoencoder), (middle) nonlinear convolutional autoencoder, and (right) lift-augmented nonlinear convolutional autoencoder.** The undistorted cases are highlighted in color while all disturbed wake cases used for training are shown with light gray curves. The lift-augmented convolutional autoencoder identifies the extreme gust–airfoil interaction manifold.

Incorporating lift into the learning process of the autoencoder assists in effectively extracting the essence of extreme aerodynamics due to three main reasons. First, using the lift force with the vorticity input comes as a natural choice since vorticity is theoretically known as the main mechanism for generating lift on a body. We take advantage of this intimate relationship between the vorticity field and lift. Second, the latent variables are guided to retain important information held by the vorticity field correlated with the lift force. This means that vortical structures that apply large vortical forces are well-captured by the lift-augmented autoencoder. Because extreme aerodynamic disturbances exert enormous amounts of transient forces as presented in Fig. 2, the autoencoder weights these extreme event appropriately and is able to accurately identify the responsible vortical structures. Third, the latent variables are encouraged to distinguish cases that yield different lift responses to impinging gust vortices. This is crucial to avoid latent variables from overlapping unnecessarily as observed in the struggling cases of PCA.

Now, let us examine the compression results from the lift-augmented autoencoder. The latent space comprised of three variables $\xi_1$, $\xi_2$, and $\xi_3$ is presented in Fig. 4. We observe that the entire collection of extreme aerodynamic cases collapses well in these latent space coordinates, confirming that the extreme aerodynamic responses to gust vortices possess a fundamentally low-dimensional behavior if captured appropriately with a nonlinear compression method. Here, the asymptotic periodic shedding states of the airfoil wakes provide shows a "cone" or a "chocolate cornet" like structure with the extreme aerodynamic trajectories lying in its vicinity in the three-dimensional latent space. As the dynamical trajectories converge to the cone-shaped structure, this structure serves as the inertial manifold[38–40]. This manifold geometry can be considered as an hour-glass shape since there is a mirrored manifold for negative angle of attack cases. It should be noted that the geometry of this structure is not specified a priori and is discovered in an unsupervised manner. Here, this surface constitutes a manifold on which the key dynamics of extreme gust response reside. That is, the trajectories of the presently considered extreme aerodynamic flows are mapped onto the discovered geometry or to its vicinity.

Given the lift-augmented autoencoder collapsing all extreme aerodynamic response data onto this cone-shaped manifold, let us examine the accuracy of state reconstruction for the disturbed flow and lift based on the three latent variables ($\xi_1, \xi_2, \xi_3$). As representative examples, we present the performance of the autoencoder for the cases of ($\alpha, G, L, y_0/c$) = (40°, −2.2, 0.5, 0.3) and (60°, −2.8, 1.5, 0), which are unused in training but chosen from the training parameter range. Here, the gust ratios $G$ for these two cases are much higher than what are traditionally considered in gust response studies, but are within the training data range of $|G| \le 4$. The latent variable trajectories for these

cases and the reconstruction of lift and vortical flows are also shown in Fig. 5. To assess the reconstruction performance, we evaluate the structural similarity index (SSIM)[41] between the reference and the decoded flow fields. The SSIM value for each decoded flow field is listed under the visualized flow reconstruction. Even the flow states exhibiting nonlinear interactions between the wing and the extreme vortex gust can be reconstructed well by the present autoencoder, as presented in Figs. 5 and S.1. Note that the structural similarity index for a regular autoencoder without lift being higher compared to that for the lift-augmented autoencoder is expected. This is because a regular autoencoder is able to tune its weights solely to obtain accurate reconstruction of the flow field from the latent variables. It is also possible to reconstruct lift solely from the vorticity field. However, the lift-augmented autoencoder is critical for revealing the manifold for extreme aerodynamic response dynamics. These successful reconstructions indicate that high-dimensional extreme aerodynamic flows can be compressed into only three variables without significant loss of key physics.

The trajectory in the present latent space for the disturbed cases reflects key features of nonlinear vortex-gust interaction appearing in the high-dimensional space. For the extreme aerodynamic case of $\alpha = 40°$, the latent vector first drops towards the direction of the undisturbed periodic orbit of $\alpha = 30°$ then comes back to the original undisturbed orbit of $\alpha = 40°$. This is due to the approach of negative vortex disturbance to the wing, decreasing the effective angle of attack. This indicates that the lift-augmented autoencoder captures the relationship between high-dimensional extreme aerodynamic flows and lift force in the low-order space. In fact, the reduction in $\xi_3$ towards the direction of $\alpha = 30°$ in the latent space coincides with the temporal evolution of lift responses, as shown in Fig. 4. A similar trend is also observed in the case of $\alpha = 60°$ in which the latent vector first heads to the direction of the periodic orbit of $\alpha = 50°$ corresponding to the decrease of the lift response. Being able to capture the extreme aerodynamic response of the wing on this manifold enables us to relate the instantaneous dynamics to the effective angle of attack, which is critically important for the flight stability of air vehicles. It is particularly encouraging that only three values in latent space are required to accurately estimate the state of the violent flow around the wing and transient lift force, which is promising for future development of sensors.

The discovered manifold captures dynamics beyond the trained gust strength $G$. Let us demonstrate how the present autoencoder approach is able to capture even more severe vortex gust conditions with $|G| \ge 4$ by presenting two cases ($\alpha, G, L, y_0/c$) = (40°, 6.4, 1.0, 0.1) and = (60°, −6.0, 2.0, −0.3), as shown in Fig. 5. The trajectory of these seemingly extrapolative cases exhibits a larger radius on the $\xi_1 - \xi_2$ plane compared to the variables of the interpolation cases while also

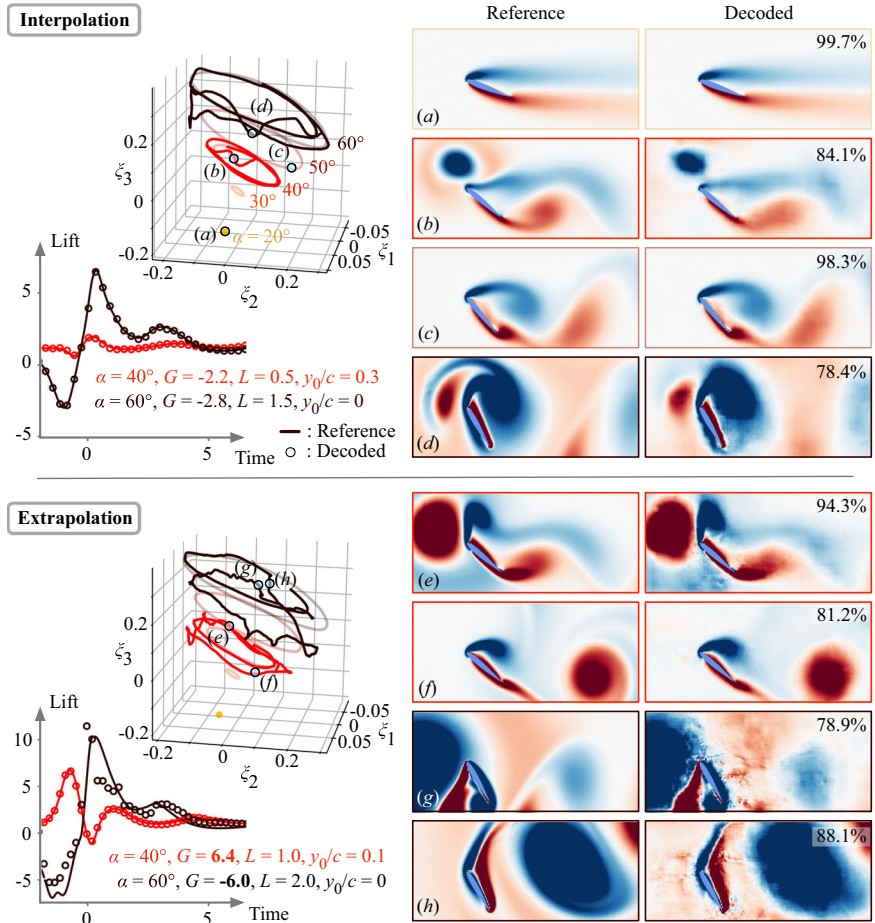

**Fig. 5 | The lift-augmented autoencoder-based manifold expression and its output for a variety of wakes around a wing.** Interpolation ($|G| < 4$) and extrapolation ($|G| > 4$) cases with regard to training data are shown. The trajectories for the undisturbed cases are shown in light color in latent space. The value presented on each decoded flow field (**a**–**h**) corresponds to the time-averaged structural similarity index between the decoded and reference fields.

presenting the effective angle of attack as the latent vector moves in the $\xi_3$ direction. The wider radial trajectory is due to the stronger disturbance, which produces a higher level of fluctuations in the vortical flow response likely away from the body without affecting lift. The present decoder can also recover the high-dimensional flow states while estimating the very large transient lift dynamics as presented in Fig. 5. This indicates that the present autoencoder can be robustly applied for such extreme gust vortex-airfoil interactions while achieving a nearly lossless compression of high-dimensional data. These findings also suggest that even under the extrapolating condition of $|G| > 4$, the underlying vortex dynamics shares common physics and can be nonlinearly compressed to a low-dimensional and universal manifold. This provides hope in estimating the flow states for cases that were not part of the extreme aerodynamic training data.

To further examine the robustness of the identified manifold and the autoencoder, let us consider cases that are different from the training cases, namely flows with noise and two extreme vortices, as shown in Fig. 6. Here, the noisy flow field is generated by adding Gaussian noise that is 30% of the original extreme aerodynamic flow field (same as the case shown in Fig. 5(*b*)). The present lift-augmented autoencoder not only reconstructs a vortical flow but also estimates the lift response well from a noisy flow. The autoencoder noise rejection characteristics is beneficial in obtaining real-time situational awareness and working with turbulent flows in which less influential smaller-scale structures may also be present around the extreme gust vortices.

We also consider cases of vortex-dominated gust flows that are challenging for most reconstruction techniques trained only with single-gust disturbances. Here, we take two vortices that are introduced vertically and horizontally upstream of the airfoil, as depicted in Fig. 6. In both cases, the dynamical lift responses can be accurately estimated while the decoder reproduces the presence of two vortex disturbances very well, as shown in Figs. 6 and S.2. We can also notice from the latent space that the trajectory of the two-horizontal-vortex case presents two inflections at $\xi_2 \approx 0$. This coincides with the observation in the lift dynamics which possesses two valleys due to the impingement of two negative vortices. We note that these particular examples are difficult for linear techniques, including PCA which completely fails to reconstruct the flow field as shown in Fig. S.2.

Finally, let us demonstrate the potential of the present lift-augmented autoencoder for handling a more challenging and realistic extreme flight condition. In this last example, we introduce randomly generated five strong vortices upstream of the wing to simulate severe wake turbulence striking the airfoil, as shown in Fig. 7 (top left). The decoded lift, reconstructed flow fields, and latent trajectory are presented in Fig. 7. Even under this extreme operating condition, the present autoencoder robustly provides accurate reconstruction of the flow variables despite the model being trained only with single-gust disturbances. This success in flow compression/reconstruction and lift estimation corroborates that the discovered low-dimensional manifold universally captures the extreme gust vortex-airfoil interaction dynamics. This discovery provides great hope in establishing flight in extreme gusts, which was traditionally considered impossible.

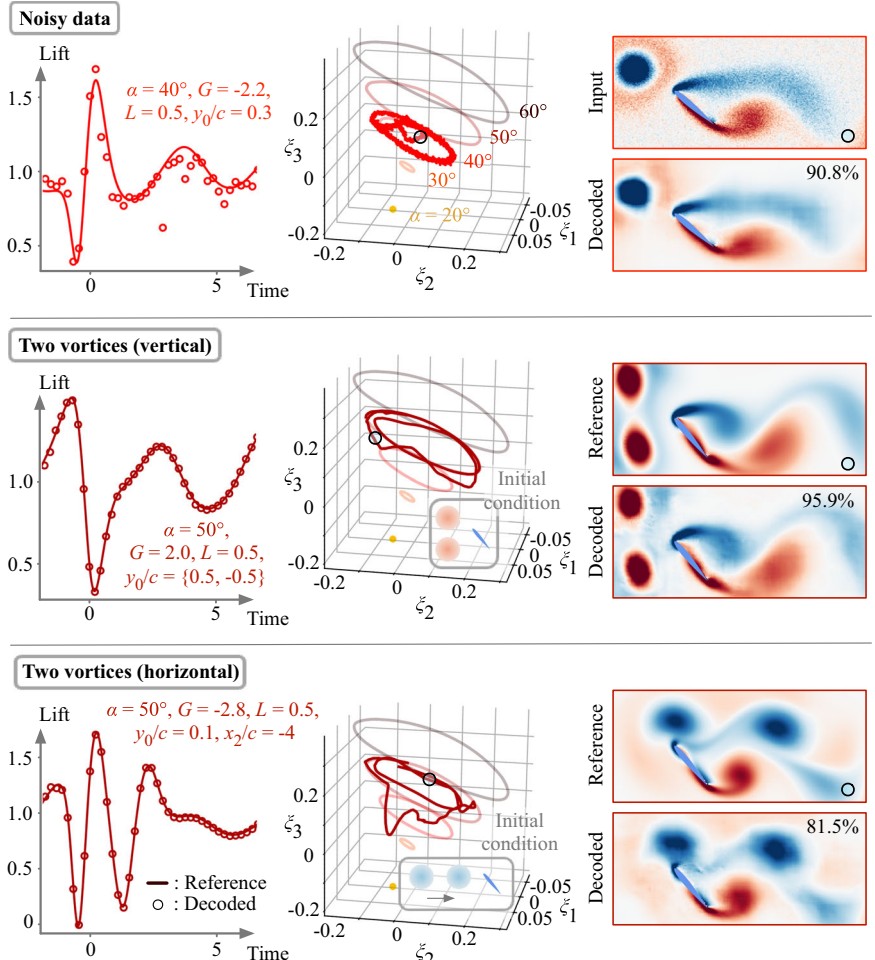

**Fig. 6 | Extrapolation assessments of the present nonlinear autoencoder with lift augmentation.** Three extrapolating cases are considered: 1. noisy data, 2. two vortex gusts (vertically arranged), and 3. two vortex gusts (horizontally arranged). The initial condition for the cases with two vortices is illustrated in the plot of each latent space. The trajectories for the undisturbed cases are shown in light color in latent space. The value presented on each decoded flow field corresponds to the time-averaged structural similarity index between the decoded and reference fields.

## Discussion

Small-scale air vehicles flying in urban or mountainous environments need to maneuver through a highly unsteady wake field full of strong vortices generated by manmade or natural obstacles. The interaction of these vortices with flying vehicles requires an understanding of extreme aerodynamic flows, for which there are no established theories. In the current study, we presented a data-driven approach to identify a low-dimensional manifold on which the key dynamics between strong gust vortices and airfoil wakes can be collapsed. This manifold was found with an autoencoder designed to retain the knowledge of aerodynamic lift as part of the latent variables. The existence of this low-dimensional manifold is significant in a few ways.

First, the fact that only three variables can represent the complex vortical flow field confirms the low-dimensional nature of the strong gust vortices interacting with the airfoil wakes. While the present study distilled the dynamics to only three variables, it actually can be further reduced to two variables if the three variables are projected on the identified manifold. This significant compression of the extreme aerodynamic flow fields was enabled with a nonlinear autoencoder-based approach that incorporates aerodynamic insights embedded into its formulation. We also note that noisy experimental data encountering a different type of gust can also be coincidentally low-dimensionalized to be three-dimensional variables through a nonlinear autoencoder with the assistance of a topology-based concept[42].

Second, this low-dimensional representation of the extreme aerodynamic flows suggests that only a small number of sensors on the airfoil may be able to accurately reconstruct the surrounding flow field in real time. In fact, decoder-type neural networks, that take sparse sensors as the input and high-resolution aerodynamic flows as the output, have been recently developed to perform real-time fluid flow state estimation[43–46]. The observations in these studies imply that low-dimensional extreme aerodynamic latent vectors can also be estimated from sparse sensor information, enabling us to track the high-dimensional dynamics in a low-order, real-time manner. Third, given the present findings, it is possible to develop a reduced-order model that can capture the dynamics in the latent space to desired level of accuracy and complexity. While we could model the latent dynamics using other data-driven techniques such as sparse regression[47], modeling and controlling the high-dimensional extreme aerodynamic flows on the present manifolds from the perspective of phase-amplitude space appears interesting[48–50]. It can be anticipated the phase-reduction analysis on the present nonlinear manifold offers a new aspect in modifying wake dynamics by providing the optimal timing and locations of actuation.

With the discovered manifold, active flow control and vehicle stability control strategies can be developed for mitigating the effects of extreme aerodynamic disturbances. While this study focused on extreme two-dimensional vortex-airfoil interactions,

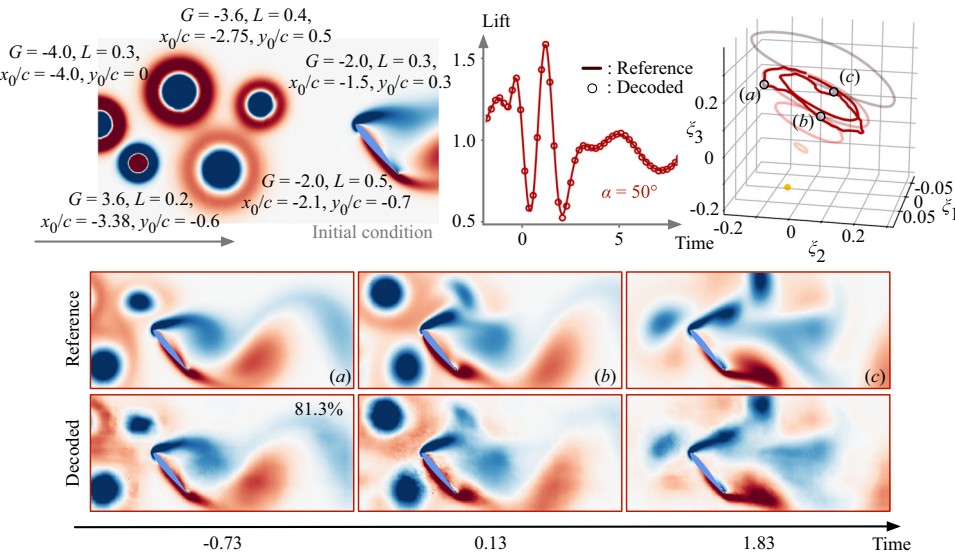

**Fig. 7 | Application of the present nonlinear autoencoder with lift augmentation to extreme gust disturbance situation with multiple vortices, simulating severe wake turbulence striking the airfoil.** The initial condition with the gust parameters for each vortex is provided (top left). The decoded lift history and the latent space dynamics are shown (top right). The trajectories for the undisturbed cases are shown in light color in latent space. The value presented on the decoded flow field (**a**–**c**) corresponds to the time-averaged structural similarity index between the decoded and reference fields (bottom).

there are other types of gust disturbances present in severe atmospheric turbulence that require three-dimensional analysis[9]. Three-dimensional flow extensions are important to further examine the potential of the present approach. As shown in this study, nonlinear data-driven compression techniques appear promising to support the identification of other manifolds that capture the complex extreme aerodynamic interactions between these other types of gusts and the aircraft. The present findings offer a new perspective on modeling and taming extreme aerodynamic flows in support of next-generation air vehicle operations in conditions traditionally considered unflyable.

## Methods

### Simulations of extreme vortex–airfoil interactions

The present study considers the unsteady flow field generated by the extreme gust vortex-airfoil interactions. The flow fields examined in this study are obtained from direct numerical simulations of flows over a NACA 0012 airfoil at a chord-based Reynolds number $\mathrm{Re} \equiv u_\infty c/\nu = 100$. Here, $u_\infty$ is the free-stream velocity, $c$ is the chord length, and $\nu$ is the kinematic viscosity. The simulations are performed with an incompressible flow solver[26,27] for an airfoil at six different angles of attack of $\alpha = 20°$, $30°$, $40°$, $50°$, and $60°$. For the undisturbed cases, the flow at $\alpha = 20°$ is steady while that at $\alpha \geq 30°$ exhibits unsteady periodic wake shedding. The computational domain extends over $(x, y)/c \in [-15, 30] \times [-20, 20]$ with the leading edge of the wing positioned at the origin.

To study the gust-airfoil interactions, a very strong vortex with an angular velocity profile prescribed by equation[28] is introduced upstream of the airfoil at $x_0/c = -2$ and $y_0/c \in [-0.5, 0.5]$. The present disturbance vortex is parameterized by the gust ratio $G \equiv u_{\theta,\max}/u_\infty \in [-10, 10]$, its size $L \equiv 2R/c \in [0.5, 2]$, and the vertical position of the disturbance $y_0/c$. From the parameter space composed of these three variables, 40 randomly-sampled cases of the disturbed flows are simulated for each angle of attack. For the purpose of learning the extreme aerodynamics with the autoencoder, 20 cases are used for training and the remaining 20 cases are used for testing. The simulated flows were validated with previous studies[45,51–53], in particular with a study that considered a vortex-airfoil interaction problem[45].

For each of the cases considered in the present study, we prepare 1200 snapshots of vorticity field over 10.2 non-dimensional convective time $t^* \equiv u_\infty t/c$. We refer to this convective time as simply 'time' in the main text. Of the entire flow field, a subdomain $(x, y)/c \in [-1.4, 4] \times [-1.2, 1.2]$ with spatial grid points $(N_x, N_y) = (240, 120)$ is considered for the data-driven analysis since vortex-airfoil interactions primarily occur in this region. Moreover, the history of the lift fluctuations is provided by the numerical simulations. The non-dimensional lift coefficient $C_L \equiv F_{\mathrm{lift}}/(\frac{1}{2}\rho u_\infty^2 c)$, where $F_{\mathrm{lift}}$ is the lift force on the wing body and $\rho$ is the density. In the main text, $C_L$ is referred to as 'lift.' Overall, the training data used for the present models amounts to $1.26 \times 10^5$ frames comprised of 100 extreme aerodynamic gust response cases and 5 undisturbed wake cases with 1200 snapshots for each case.

### Autoencoder setup

To discover the universal nonlinear manifold that represents the high-dimensional extreme aerodynamic flows in a low-dimensional latent space, we use an autoencoder[33] (see Fig. 3). Here, we consider a convolutional neural-network-based autoencoder $\mathcal{F}$, which is trained to output $\hat{q}$ to be the same data as the input $q \in \mathbb{R}^n$ such that $\hat{q} \approx \mathcal{F}(q; w)$, where $w$ denotes the weights inside the autoencoder. This autoencoder is comprised of an encoder $\mathcal{F}_e$ and a decoder $\mathcal{F}_d$ connected through a low-dimensional variable $\xi \in \mathbb{R}^m$ in the middle, where $m \ll n$. Here, the high-dimensional input $q$ can be compressed into the latent vector $\xi$ if the autoencoder $\mathcal{F}$ successfully recovers the data accurately. That is, we seek to have an autoencoder that achieves

$$q \approx \hat{q} = \mathcal{F}(q; w) = \mathcal{F}_d(\xi) = \mathcal{F}_d(\mathcal{F}_e(q)). \tag{1}$$

The autoencoder $\mathcal{F}$ is found based on data such that its weights $w$ are optimized to minimize a desired cost (loss) function $\mathcal{E}$, yielding the following optimization problem

$$w = \mathrm{argmin}_w[\mathcal{E}(q, \mathcal{F}(q; w))] = \mathrm{argmin}_w \| q - \hat{q} \|_2. \tag{2}$$

The weights $w$ are determined with the Adam optimizer[54].

We use an autoencoder composed of convolutional neural networks (CNN)[36] and multi-layer perceptrons (MLP)[37], as illustrated in

Fig. 3. In the encoder, the CNN captures global features of the extreme aerodynamic flow field and the MLP is used to extract features from the CNN while further reducing the size of the data. By leveraging nonlinear activation functions, an autoencoder can achieve better compression than linear compression techniques such as principal component analysis (PCA)[33]. Note that using autoencoder with linear activation functions is mathematically equivalent to performing principal component analysis (PCA)[33,34]. As for the nonlinear activation function, we use the hyperbolic tangent function $\varphi(s) = (e^s - e^{-s})/(e^s + e^{-s})$, enabling us to consider the positive and the negative gust influence in latent space. The hyperparameters used in the MLP and CNN follow previous work with similar settings[34,55]. Full details on the parameters of the present convolutional nonlinear autoencoder are shown in table S.1.

In addition to PCA and a regular autoencoder, we develop a lift-augmented convolutional autoencoder in this study. The present lift-augmented autoencoder trains the model with a lift coefficient $C_L(t)$ in addition to a vorticity field $\boldsymbol{q}(t)$ such that $[\hat{\boldsymbol{q}}(t), \hat{C}_L(t)] = \mathcal{F}(\boldsymbol{q}(t))$. The additional side network based on an MLP is illustrated in the green-shaded portion of Fig. 3. This additional network ensures that the latent vector $\boldsymbol{\xi}(t)$ holds relevant information related to the lift coefficient $C_L(t)$ to support the manifold identification. The cost function in this case becomes

$$\boldsymbol{w}^* = \text{argmin}_{\boldsymbol{w}}\left[||\boldsymbol{q} - \hat{\boldsymbol{q}}||_2 + \beta||C_L - \hat{C}_L||_2\right], \quad (3)$$

where $\beta$ balances the vorticity field and lift reconstruction losses. In this study, we choose $\beta = 0.05$ based on the L-curve analysis[56]. With the lift decoder $\mathcal{F}_L$, the reconstructed lift coefficient $\hat{C}_L(t)$ is given by

$$\hat{C}_L(t) = \mathcal{F}_L(\boldsymbol{\xi}(t)) = \mathcal{F}_L(\mathcal{F}_e(\boldsymbol{q}(t))). \quad (4)$$

## Data availability
Extreme aerodynamic data[57] used in the present study are available on the Open Science Framework (https://doi.org/10.17605/OSF.IO/7VSH8). Source data are provided with this paper, which is also available on the same link above.

## Code availability
Sample codes[58] for training the present models are available on GitHub (https://github.com/kfukami/Observable-AE and https://zenodo.org/badge/latestdoi/677509021).

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

## Acknowledgements

We thank A. Linot, H. Nakao, and L. Smith for stimulating discussions on nonlinear dynamics and compression. Support for illustrations was provided by T. Taira. K.T. acknowledges the generous support from the US Department of Defense Vannevar Bush Faculty Fellowship (grant number: N00014-22-1-2798) and the US Air Force Office of Scientific Research (grant number: FA9550-21-1-0178). K.F. acknowledges the support from the UCLA-Amazon Science Hub for Humanity and Artificial Intelligence.

## Author contributions

K.T. conceptualized the approach. K.F. and K.T. selected and analyzed the problems and wrote the manuscript. K.F. developed the software, curated the data, and visualized the results. K.T. secured funding support and supervised the project.

## Competing interests

The authors declare no competing interests.
