## [Peer Review File · Nature Communications]

Grasping extreme aerodynamics on a low-dimensional manifoldREVIEWER COMMENTS

Reviewer #1 (Remarks to the Author):

Main comments:

This work focuses on modeling extreme unsteady gusty flows, specifically addressing the challenges associated with modeling extreme disturbance vortices due to their strong nonlinearity. Different from traditional modeling methods, the paper proposes a reduction modeling approach using a nonlinear autoencoder. It reduces the temporal frames into a latent space with 3 dimensions. Surprisingly, despite the low dimensionality, this latent space is capable of effectively reconstructing data under various conditions such as different airfoil angles and measurement noise. Moreover, it demonstrates generalizability under complex scenarios involving multiple vortices. In general, this paper provides a new perspective for complex system modeling, and could contribute to the control of air vehicles under this type of extreme environment.

This work is expected to have a certain level of significance to the field. Autoencoders is designed for reduction modeling for various complex (especially nonlinear) systems. The reviewer believes the methodology provided in this paper is sound. Autoencoders are generally possible to perform reduced-order modeling (proven in many tasks), as demonstrated in [1]. By tackling the modeling of extreme unsteady gusty flows with autoencoders, this paper offers valuable insights that could benefit researchers and practitioners working in the field of fluid dynamics and aerodynamics. This work is likely to be of interest to a wide range of researchers.

In terms of the modeling/control of UAVs under extreme gusty flows, this work has the potential to open up the field of extreme gusty flow control. One of the baseline deep learning model is similar to prior works like [2] (cited in draft). Part of this work further extends the study of [2] to extreme gust vortices.

All the experiments are well demonstrated with details to support the claims. The analysis and interpretation is clean. The experiments are thoughtfully designed, covering various interesting aspects of the problem at hand.

In the following, the reviewer aims to point out certain concerns and minor flaws identified in the paper. The reviewer intends to address these questions to the authors in order to seek further clarification.

- i) [Question] The experimental study is only demonstrated in 2D cases. Would the authors plan to build more experiments in 3D for future work? How would this expect to cost computationally?
- ii) [Suggestion] This paper is mostly using simulation data. What would be the difficulty/gap when one tries to apply this method to real data? Would it cost the required latent dimension to be more than three? Moreover, this paper uniformly suggests a model with latent dimension=3. It might be better to write a few lines to discuss the reason.
- iii) [Discussion] AE with lift is acceptable. Strictly speaking, AE with lift should be compared to AE without lift using latent dimension=4 (3 original latent dimensions + 1 lift). This would create a more cohesive modeling approach. However, the reviewer acknowledges that such a modeling might pose challenges for interpretation. The reviewer would appreciate some further discussion from the authors regarding this particular point.

Minor points:

- i) [Question] For the reconstruction percentage in Fig. 5, 6, 7, do they represent the reconstruction rate of this specific snapshot, or is it averaged over all the snapshots?
- ii) [Discussion] In Fig. 8, the performance of AE with lift slightly underperforms AE without lift.

Maybe some very 'advanced' hyper-parameter tuning could help to fix the problem?

iii) In abstract, there should be a 'will/would' in front of the first 'encounter'.

[1] Chen, Boyuan, et al. "Automated discovery of fundamental variables hidden in experimental data." *Nature Computational Science* 2.7 (2022): 433-442.

[2] Omata, Noriyasu, and Susumu Shirayama. "A novel method of low-dimensional representation for temporal behavior of flow fields using deep autoencoder." *Aip Advances* 9.1 (2019): 015006.

Reviewer #2 (Remarks to the Author):

The manuscript entitled "Grasping Extreme Aerodynamics on a Low-Dimensional Manifold" by Kai Fukami and Kunihiro Taira presents a data-driven approach to identify a low-dimensional manifold on which the key dynamics between strong gust vortices and airfoil wakes can be collapsed. As also stated by the authors, the interaction between the vortical gusts and wings is seemingly complex and different for each combination of gust parameters, however they achieved to discover a manifold with an autoencoder designed to retain the knowledge of aerodynamic lift as part of the latent variables and showed that the fundamental physics behind extreme aerodynamics is far simpler and low-rank than traditionally expected. It is a very interesting, significant work, a valuable research presented in a clear and complete form. Therefore, I would recommend that the paper be published as is. However, I would like to point out a couple of concerns for the perfectness of the manuscript.

If I understand correctly, ξ_3 is related to the effective angle of attack and the width of the radial trajectory on the ξ_1 and ξ_2 plane (combination of ξ_1 and ξ_2) shows the strength of the disturbance. I assume the effective angle of attack will depend on all of the parameters of the study, namely α , G , L , and y_0/c . On the other hand the strength of the disturbance or its consequence in terms of lift force depends on all of the parameters of the study as well. Therefore, I expect the latent variables are independent and represents a physical and probably a non-dimensional value made of a combination of those parameters. It would be very insightful if the authors could also comment on how physically the latent variables are related to the parameters of the study. According to the results presented in Figure S1, structural similarity indices for AE with lift is a bit smaller than those for AE w/o lift. A brief explanation for the reason behind will be helpful to understand the approach and the flow physics. Having noticed this slight difference, I wonder if the authors obtained the lift from the reconstructed flow field data with the regular autoencoder and compared with that obtained with lift-augmented autoencoder. And finally, I believe a measure for the accuracy of the lift may support the success of the discovered manifold. In terms of the structure of the manuscript, I would like to point out only that in the Supplemental Material, the references go up to #35 while in the reference list of the main text, the final reference is numbered as #30.

Reviewer #3 (Remarks to the Author):

The paper presents findings that offer a new perspective on modeling gusts using machine learning. It has interesting results and utilizes multi-disciplinary research to achieve them. I believe the scoping of the paper needs further work and the big claims made need to be toned down a little. The paper will benefit focusing on the ability to reduce the problem to 3 parameters.

The abstract and introduction don't introduce the content of the paper properly or clearly. For example, the application of the work is explained at end of paper whereby the low dimensional representation can help the use of sensors on an aircraft. I think this needs to be highlighted at the start of the paper for context. But also note that there has already been work on the use of ML

to predict stall over a 3D wing by Shane Windsor in University of Bristol and others. Discussing this early will set the scene and enable the reader to follow the vision of why this work is important and how it can be used to improve safety. There are some bold claims which I don't agree with entirely such as "However, there exists virtually no foundation to describe the influence of extreme vortical gusts on flying bodies". I think the foundation through CFD simulation and experimental work does exist and the review papers by Anya Jones list them. There has even been attempts to explore worst case gusts in urban environments by simulating flight through shear layers which are far more "extreme" than the case being considered in this paper:

- Mohamed, A.; Marino, M.; Watkins, S.; Jaworski, J.; Jones, A. Gusts Encountered by Flying Vehicles in Proximity to Buildings. *Drones* 2023, 7, 22. <https://doi.org/10.3390/drones7010022>

This gust has even been characterized as extreme in this publication:

- Colin M. Stutz, John T. Hryniuk, Douglas G. Bohl, Dimensional analysis of a transverse gust encounter, *Aerospace Science and Technology*, Volume 137, 2023, 108285, ISSN 1270-9638, <https://doi.org/10.1016/j.ast.2023.108285>.

I suggest the authors somehow explain the scope of the gusts ($G > 1$) they will explore earlier in the manuscript in a clearer way. I also suggest avoiding the term "extreme aerodynamics" which seems subjective and is not representative since it doesn't cite prior work (example Colin et al). I think just calling it a strong gust is sufficient just like the authors already did in Paragraph 1 of Extreme Vortex Airfoil Interactions section whereby it was called strong.

More specific comments below:

Introduction Para2: It would be good to acknowledge that UAVs already operate to deliver parcels for example in Australia by SwoopAero, and Google X since 2019. Therefore statements like this needs to be updated since the technology has already started operation, "These new and amazing flying vehicle concepts will likely revolutionize air-based transportation5, 10, 11".

Introduction Para3: Suggest to not list gusts and vortical disturbances as two separate things. It may be good to reference the gust taxonomy and use the terminology from it:

- Mohamed, A.; Marino, M.; Watkins, S.; Jaworski, J.; Jones, A. Gusts Encountered by Flying Vehicles in Proximity to Buildings. *Drones* 2023, 7, 22. <https://doi.org/10.3390/drones7010022>

Also the statement of "Such a flight environment has been off-limits due to the fact that there is virtually no available theory for extreme aerodynamic problems and to avoid possible loss of aircraft", is not accurate since drones have operated in these environments (see references below).

- Mohamed, A., Abdulrahim, M., Watkins, S., & Clothier, R. (2016). Development and flight testing of a turbulence mitigation system for micro air vehicles. *Journal of Field Robotics*, 33(5), 639-660.

- Prudden, S., Fisher, A., Marino, M., Mohamed, A., Watkins, S., & Wild, G. (2018). Measuring wind with small unmanned aircraft systems. *Journal of Wind Engineering and Industrial Aerodynamics*, 176, 197-210.

- Gavrilovic, N., Mohamed, A., Marino, M., Watkins, S., Moschetta, J. M., & Bénard, E. (2018). Avian-inspired energy-harvesting from atmospheric phenomena for small UAVs. *Bioinspiration & biomimetics*, 14(1), 016006.

Also you don't need to understand the theory for a controller to mitigate the disturbances which has already been demonstrated for fixed and rotary wings... As mentioned before researchers are even exploring the use of ML to create controllers that use an array of sensors during stall (see below)... The narrative needs to therefore be scoped down and more focused.

- Araujo-Estrada, S. A., & Windsor, S. P. (2021). Aerodynamic state and loads estimation using bioinspired distributed sensing. *Journal of Aircraft*, 58(4), 704-716.

Introduction Para6 / last sentence: Based on the fact that the focus of the paper is on use of ML to only model a vortex hitting a canonical airfoil as stated in this sentence; its best to rescope the abstract and introduction accordingly since you only considered 1 orientation of flight through a

vortex in a 2D sense. There are other flow disturbances (such as shear layers) and even flying through them at different orientations/angles which is important from a 3D perspective that you have not yet explored. There is also the fact that experiencing a single discrete vortex can be different to experiencing a series of vortices of different scales (more realistic) which is probably worth mentioning as a condition to consider in the future.

Figure1 (Right) a scale is missing.

Extreme Vortex Airfoil interaction Para1: Can you add any validation of the simulations undertaken to increase confidence in the simulations. I couldn't find it in the Supp material too. Comparing the simulated flow behavior to published experiments is important here.

Figure2: What is unit of Lift on y-axis? Can it be non-dimensionalised? Also Im not sure what all the light colored curves are on the plot? Finally I assume that the contour represents pressure? Its stated in figure.

Extreme Vortex Airfoil interaction Para3: It would be good to explicitly state the RE here. Also worth noting that atmospheric flow is rarely laminar which affects flow behavior significantly. Acknowledging this is important as part of stating the underlying assumptions and explaining it's a step towards more complex and realistic simulations.

The descriptions of the reconstructed lift isn't very clear. Does this mean that the state variables are used to produce the lift value given the latent state values? If so, why isn't lift just another part of the auto-encoder input/output? Why is the augmentation required?

"autoencoder to compress the vorticity field to mere three variables is not only surprising but also reaffirms that the flow field is indeed comprised of common flow features" → not really surprising given autoencoder literature - the whole point is to use few features! the PCA analysis' highlight few field being used.

Images start switching "input" and "reference" image in the figures, but doesn't clearly describe this distinction in the text. Is the reference input? or is this what should be being produced by the network? if so, where does this reference image come from?

I note these can mean different things depending on the context

Is this just showing the input and output can be replicated? Why is this significant for nature? How this this actually applied?

Reviewer #4 (Remarks to the Author):

This paper reveals that extreme aerodynamic fields can be beautifully embedded in low-dimensional space by data-driven methods.

The idea of introducing a lift-augmented structure to the traditional autoencoder is simple and clear: the features are left with enough information to estimate the lift. However, the difference emerged by this modification is significant, and the feature space that the autoencoder composes becomes completely different. In particular, it is surprising that it can be generalized to extrapolative situations such as those shown in the lower part of Figure 5, Figure 6, and Figure 7. The description of the paper is basically clear and well-organized.

It would be desirable to write more clearly what is to be done with the obtained feature space and why we are happy to have this embedding throughout the text.

Several interesting expected specific applications are given in the discussion section, e.g:

"low-dimensional representation of the extreme aerodynamic flows suggests that only a small number of sensors on the airfoil may be able to accurately reconstruct the surrounding flow field in real time",

"it is possible to develop a reduced-order model that can capture the dynamics in the latent space to desired level of accuracy and complexity",

"With the discovered manifold, active flow control and vehicle stability control strategies can be developed for mitigating the effects of extreme aerodynamic disturbances".

It would be helpful for readers to additionally explain the connection between the applications and the proposed method. It would be even better if this explanation could be related to existing works including the authors' ones dealing with flow field data based on autoencoders.

Other more specific comments are as follows:

While the title and other text uses word "manifold", can the destination of this embedding be considered as a manifold? It seems to me that it is just a "space" of three dimensions. Are you saying that 3-dimensional \mathbb{R}^3 space is a manifold in some sense? Or are you referring to the cone-shaped structure? If so, is the structure really within 2 dimensions (even approximately)? From the right side of Fig. 4, it seems to be widely distributed in the three-dimensional space. The claim in the discussion section that the embedding can be reduced to two variables is also questionable.

In the abstract, it is stated that "there is an enormous parameter space for gusty conditions wings encounter". What does this mean? I could not find a detailed description in the main text. Do you mean the space of four parameters (or, plus Re) in the experiment? Or do you mean a much larger space of parameters? If the latter, is 4 or 5 dimensions in the authors' experiment sufficient?

The abstract states that "the fundamental physics behind extreme aerodynamics is far simpler and low-rank than traditionally expected". How much dimensionality has conventionally expected by whom? It should be stated in the main text.

The Reynolds number used in this study is fixed at 100. What specific application is this Reynolds number set with in mind? Is it not too low?

The following sentences in page 3:

"The current choice of Re captures flows that are representative of a wide range of realistic flow scenarios since the extreme aerodynamic disturbance creates high local flow acceleration injecting a large amount of vorticity from the surface. This generally leads to laminarizing the flow near the body."

may be in response to this question, but if so, could you provide more detailed explanation?

In page 5, the authors stated that "While powerful at compressing Gaussian-distributed data, PCA is known to have difficulty in compressing transient physics and those with strong nonlinearities that produce non-Gaussian distributions, which is certainly the case with extreme aerodynamic flows", which seems unclear to me.

It is true that PCA can be regarded as a probabilistic modeling with multidimensional normal distribution, but, in my understanding, Gaussianity and the compression performance is almost unrelated. What determines the compression rate by PCA is how much data lies in a limited linear subspace. The reason AE works is that this subspace can be bent by nonlinear transformations. In addition, I don't understand well the term "nonlinearity". The meaning of the term nonlinear seems unclear throughout the main text. What is nonlinear object in this sentence, for example? Do you mean that the Navier-Stokes equation is a nonlinear differential equation? Is it OK to discuss the linearity/nonlinearity of PCA/AE transformations and the nonlinearity of differential equations as the same thing?

Figures 5, 6 and 7 show the structural similarity index. Since it is not a general index value, it should be mentioned briefly in the text.

In page 7, the authors stated that "it can also be considered as an hour-glass shape since there is a mirrored manifold for negative angle of attack cases."

Since AE performs nonlinear transformations, I don't think that simply mirroring the geometry works in general. Have you tried embedding the negative angle of attack cases? If it works, it is rather interesting result and should be further discussed.

Response to Reviewer 1

Journal: Nature Communications (Manuscript ID: NCOMMS-23-21038)

Title: Grasping Extreme Aerodynamics on a Low-Dimensional Manifold

Authors: Kai Fukami, Kunihiro Taira

Date sent: August 12, 2023

The authors would like to thank the referees for taking his/her time to review our manuscript. Based on the reviewers' insightful comments, revisions have been made to the manuscript, which are highlighted in red. We hope our responses fully address the comments from the referee. In what follows, we list the referee's comments with our responses shown in blue.

.....

This work focuses on modeling extreme unsteady gusty flows, specifically addressing the challenges associated with modeling extreme disturbance vortices due to their strong nonlinearity. Different from traditional modeling methods, the paper proposes a reduction modeling approach using a nonlinear autoencoder. It reduces the temporal frames into a latent space with 3 dimensions. Surprisingly, despite the low dimensionality, this latent space is capable of effectively reconstructing data under various conditions such as different airfoil angles and measurement noise. Moreover, it demonstrates generalizability under complex scenarios involving multiple vortices. In general, this paper provides a new perspective for complex system modeling, and could contribute to the control of air vehicles under this type of extreme environment.

This work is expected to have a certain level of significance to the field. Autoencoders is designed for reduction modeling for various complex (especially nonlinear) systems. The reviewer believes the methodology provided in this paper is sound. Autoencoders are generally possible to perform reduced-order modeling (proven in many tasks), as demonstrated in [1]. By tackling the modeling of extreme unsteady gusty flows with autoencoders, this paper offers valuable insights that could benefit researchers and practitioners working in the field of fluid dynamics and aerodynamics. This work is likely to be of interest to a wide range of researchers.

In terms of the modeling/control of UAVs under extreme gusty flows, this work has the potential to open up the field of extreme gusty flow control. One of the baseline deep learning model is similar to prior works like [2] (cited in draft). Part of this work further extends the study of [2] to extreme gust vortices.

All the experiments are well demonstrated with details to support the claims. The analysis and interpretation is clean. The experiments are thoughtfully designed, covering various interesting aspects of the problem at hand. In the following, the reviewer aims to point out certain concerns and minor flaws identified in the paper. The reviewer intends to address these questions to the authors in order to seek further clarification.

Response: Thank you for the encouraging remarks. We have revised our manuscript based on your comments and hope that the updated manuscript addresses your comments.

i) [Question] The experimental study is only demonstrated in 2D cases. Would the authors plan to build more experiments in 3D for future work? How would this expect to cost computationally?

Response: Yes, the extension to three-dimensional flow fields is definitely one of our upcoming tasks. In fact, we are preparing for the public an open-access data set of three-dimensional vortical flows around a wing with a good number of configurations covering a variety of wake patterns (Ribeiro et al., arXiv preprint, 2023). We are very excited about extending the proposed method for such three-dimensional flows.

From the aspect of model formulation, applications to three-dimensional flows can be achieved by solely replacing the two-dimensional convolutional operations with three-dimensional operations. Regarding the computational cost, it would be more expensive compared to the present two-dimensional cases. However, it will still be manageable because the convolutional operations in the network greatly reduce the computational cost compared to the fully-connected models. We now mention extensions for three-dimensional flows in the third paragraph of Discussions on page 10. We also note that our group has extensive experience with two-

and three-dimensional machine-learning-based super-resolution analysis that will help with extending our analysis to three-dimensional flows (Fukami et al., Theor. Comput. Fluid Dyn., 2023).

- Ribeiro, J. H. M., Neal, J., Burtsev, A., Amitay, M., Theofilis, V., & Taira, K. (2023). Laminar post-stall wakes of tapered swept wings. arXiv:2304.07587.
- Fukami, K., Fukagata, K., & Taira, K. (2023). Super-resolution analysis via machine learning: a survey for fluid flows. Theoretical and Computational Fluid Dynamics, 1-24.

ii) [Suggestion] This paper is mostly using simulation data. What would be the difficulty/gap when one tries to apply this method to real data? Would it cost the required latent dimension to be more than three? Moreover, this paper uniformly suggests a model with latent dimension = 3. It might be better to write a few lines to discuss the reason.

Response: This is a great point. When we apply the proposed approach to real data, for example, measured from experiments, one challenge is how to deal with noise in the measured vortical flow data. However, it is generally known that nonlinear autoencoders are able to extract key features of high-dimensional data in a low-order manner while performing denoising. In fact, we have recently demonstrated for noisy experimental data that a nonlinear autoencoder with the assistance of a topology-based concept can reveal the low-dimensional manifold of high-dimensional aerodynamic flows with transient gust encounter (which is a different type of gust; Smith et al., 2023). The low-dimensional latent space for such a flow is coincidentally rank 3.

The number of the latent vector required to capture the dynamics is generally dependent on the variety of flow features held in the data set, more than the level of noise contained therein. Hence, the handling of experimental or computational data does not seem to alter our findings in any substantial manner. For the present study, three latent variables are needed to express the amplitude, timing, and effective angle of attack, in high-dimensional extreme aerodynamic flows. These parameters lead to the emergence of certain flow features. We have now added this description to the second paragraph of the Discussions.

- Smith, L., Fukami, K., Sedky, G., Jones, A., & Taira, K. (2023). A cyclic perspective on transient gust encounters through the lens of persistent homology. arXiv:2306.15829.

iii) [Discussion] AE with lift is acceptable. Strictly speaking, AE with lift should be compared to AE without lift using latent dimension= 4 (3 original latent dimensions + 1 lift). This would create a more cohesive modeling approach. However, the reviewer acknowledges that such a modeling might pose challenges for interpretation. The reviewer would appreciate some further discussion from the authors regarding this particular point.

Response: The comparison between the present lift-augmented autoencoder with 3 latent variables and a regular autoencoder with 4 latent variables is actually not appropriate because the lift augmentation does not correspond to the addition of the latent dimension. To discuss how manifold identification can be promoted with the same latent dimension, the comparison should be performed with a regular autoencoder with 3 latent variables. In fact, a regular autoencoder slightly outperforms the lift-augmented autoencoder in terms of vortical flow reconstruction, as shown in figure S1, although their latent variable distribution is not physically interpretable.

Minor points:

i) [Question] For the reconstruction percentage in Fig. 5, 6, 7, do they represent the reconstruction rate of this specific snapshot, or is it averaged over all the snapshots?

Response: They are time-averaged values over all snapshots. We have revised the text to clarify this point. Thank you.

ii) [Discussion] In Fig. 8, the performance of AE with lift slightly underperforms AE without lift. Maybe some very 'advanced' hyper-parameter tuning could help to fix the problem?

Response: Related to comment iii) above, this is expected because the decoded vorticity field is only a function of latent variable $\xi \in \mathbb{R}^3$ (and not a function of C_L). Again, the present augmentation with C_L does not correspond to the addition of the latent dimension. Since the weighting for the reconstruction loss of the lift-augmented autoencoder becomes smaller compared to a regular autoencoder (see equation 3 in supplemental material), the reconstructed vorticity field from a regular autoencoder is generally better than that from the lift-augmented autoencoder.

iii) In abstract, there should be a 'will/would' in front of the first 'encounter'.

Response: Thank you. We have added "would" in the appropriate location.

Response to Reviewer 2

Journal: Nature Communications (Manuscript ID: NCOMMS-23-21038)

Title: Grasping Extreme Aerodynamics on a Low-Dimensional Manifold

Authors: Kai Fukami, Kunihiko Taira

Date sent: August 12, 2023

The authors would like to thank the referees for taking his/her time to review our manuscript. Based on the reviewers' insightful comments, revisions have been made to the manuscript, which are highlighted in red. We hope our responses fully address the comments from the referee. In what follows, we list the referee's comments with our responses shown in blue.

.....

The manuscript entitled "Grasping Extreme Aerodynamics on a Low-Dimensional Manifold" by Kai Fukami and Kunihiko Taira presents a data-driven approach to identify a low-dimensional manifold on which the key dynamics between strong gust vortices and airfoil wakes can be collapsed. As also stated by the authors, the interaction between the vortical gusts and wings is seemingly complex and different for each combination of gust parameters, however they achieved to discover a manifold with an autoencoder designed to retain the knowledge of aerodynamic lift as part of the latent variables and showed that the fundamental physics behind extreme aerodynamics is far simpler and low-rank than traditionally expected. It is a very interesting, significant work, a valuable research presented in a clear and complete form. Therefore, I would recommend that the paper be published as is. However, I would like to point out a couple of concerns for the perfectness of the manuscript.

Response: Thank you for the encouraging remarks. We have revised our manuscript based on your comments and hope that the updated manuscript addresses your comments.

1. If I understand correctly, ξ_3 is related to the effective angle of attack and the width of the radial trajectory on the ξ_1 and ξ_2 plane (combination of ξ_1 and ξ_2) shows the strength of the disturbance. I assume the effective angle of attack will depend on all of the parameters of the study, namely α , G , L , and y_0/c . On the other hand the strength of the disturbance or its consequence in terms of lift force depends on all of the parameters of the study as well. Therefore, I expect the latent variables are independent and represents a physical and probably a non-dimensional value made of a combination of those parameters. It would be very insightful if the authors could also comment on how physically the latent variables are related to the parameters of the study.

Response: This is a great comment. The parameters mentioned above have an implicit influence on the latent variables. The influence of the parameters on the flow fields is not as clear as what may be expected, as commented by the referee. We have actually considered this point for a separate study of turbulent flows (Fukami, Goto, and Taira, 2023). Within the context of the present paper, we have not commented on this point in detail since the compression of the flow field data examined here is based on the flow features and not directly on the parameters. We hope to make a stronger case for examining the connection between the problem parameters and the flow field as we extend the separate study in preparation. Please stay tuned.

- Fukami, K., Goto, S., & Taira, K. (2023). Assessing inter- and extrapolation in learning turbulence by data-driven Buckingham Pi scaling. to be submitted.

2. According to the results presented in Figure S1, structural similarity indices for AE with lift is a bit smaller than those for AE w/o lift. A brief explanation for the reason behind will be helpful to understand the approach and the flow physics. Having noticed this slight difference, I wonder if the authors obtained the lift from the reconstructed flow field data with the regular autoencoder and compared with that obtained with lift-augmented autoencoder. And finally, I believe a measure for the accuracy of the lift may support

the success of the discovered manifold.

Response: The structural similarity indices for AE without lift being higher for the results from an AE with lift is expected. This is because the AE without lift is able to solely tune its weights to optimize accurate reconstruction of the flow field from the latent variables.

It is also possible to reconstruct lift directly from the vorticity field. However, let us emphasize that the lift-augmented AE is utilized to reveal the manifold for extreme aerodynamic responses, more so than estimating the lift (although that aspect is important as well).

We appreciate the comment on the connection between the accuracy of lift estimation and the success of discovering the manifold. This is certainly true and is incorporated into the formulation by how we select the parameter β in the loss function. We have added these descriptions to the fourth paragraph on page 7.

3. In terms of the structure of the manuscript, I would like to point out only that in the Supplemental Material, the references go up to #35 while in the reference list of the main text, the final reference is numbered as #30.

Response: We have made appropriate corrections. Thank you.

Response to Reviewer 3

Journal: Nature Communications (Manuscript ID: NCOMMS-23-21038)

Title: Grasping Extreme Aerodynamics on a Low-Dimensional Manifold

Authors: Kai Fukami, Kunihiro Taira

Date sent: August 12, 2023

The authors would like to thank the referees for taking his/her time to review our manuscript. Based on the reviewers' insightful comments, revisions have been made to the manuscript, which are highlighted in red. We hope our responses fully address the comments from the referee. In what follows, we list the referee's comments with our responses shown in blue.

.....

The paper presents findings that offer a new perspective on modeling gusts using machine learning. It has interesting results and utilizes multi-disciplinary research to achieve them. I believe the scoping of the paper needs further work and the big claims made need to be toned down a little. The paper will benefit focusing on the ability to reduce the problem to 3 parameters.

Response: Thank you for very careful review and valuable comments. We have revised our manuscript based on your comments and hope that the updated manuscript addresses your comments.

1. The abstract and introduction don't introduce the content of the paper properly or clearly. For example, the application of the work is explained at end of paper whereby the low dimensional representation can help the use of sensors on an aircraft. I think this needs to be highlighted at the start of the paper for context. But also note that there has already been work on the use of ML to predict stall over a 3D wing by Shane Windsor in University of Bristol and others. Discussing this early will set the scene and enable the reader to follow the vision of why this work is important and how it can be used to improve safety.

Response: Thank you for the comment. We have now highlighted how the discovered manifold can be leveraged for not only understanding extreme aerodynamic flows but also aiding downstream tasks such as real-time flow estimation, dynamical modeling, flow control, and vehicle design, throughout the manuscript. In addition to the abstract and the introduction, we also have additional discussions in the last section of the paper to present possible future directions based on our findings. We already started a few efforts on the aforementioned items and have achieved promising results. These ongoing works will appear soon — please stay tuned.

We also thank the referee for mentioning a study by Araujo-Estrada & Windsor (2021). We feel that it is important to cite their work and have now included it in the text.

2. There are some bold claims which I don't agree with entirely such as "However, there exists virtually no foundation to describe the influence of extreme vortical gusts on flying bodies". I think the foundation through CFD simulation and experimental work does exist and the review papers by Anya Jones list them. There has even been attempts to explore worst case gusts in urban environments by simulating flight through shear layers which are far more "extreme" than the case being considered in this paper:

- Mohamed, A.; Marino, M.; Watkins, S.; Jaworski, J.; Jones, A. Gusts Encountered by Flying Vehicles in Proximity to Buildings. *Drones* 2023, 7, 22. <https://doi.org/10.3390/drones7010022>

This gust has even been characterized as extreme in this publication:

- Colin M. Stutz, John T. Hrynyuk, Douglas G. Bohl, Dimensional analysis of a transverse gust encounter, *Aerospace Science and Technology*, Volume 137, 2023, 108285, ISSN 1270-9638, <https://doi.org/10.1016/j.ast.2023.108285>

Response: Thank you for the comments. We should have been more explicit as there are some empirical data available. The mentioned statement now reads “However, there exists virtually no theoretical fluid-dynamic foundation to describe the influence of extreme vortical gusts on flying bodies.” We also have made sure to cite the aforementioned papers in the revised text.

3. I suggest the authors somehow explain the scope of the gusts ($G > 1$) they will explore earlier in the manuscript in a clearer way. I also suggest avoiding the term “extreme aerodynamics” which seems subjective and is not representative since it doesn’t cite prior work (example Colin et al). I think just calling it a strong gust is sufficient just like the authors already did in Paragraph1 of Extreme Vortex Airfoil Interactions section whereby it was called strong.

Response: This choice of the phrase “extreme aerodynamics” was not made lightly. We have observed that linear control and modeling techniques perform somewhat well up to $G \approx 1$ (despite the fact that the flow fields were not predicted well, which speaks to the robustness of those techniques). After a few of years of us having extensive discussions with stakeholders in unsteady aerodynamics, we have chosen to distinguish very strong disturbances that make these control and modeling techniques obsolete. The standard description of “strong gust” does not fully capture when the influence of gust becomes too strong. For these reasons, our colleagues in the academia, industry, and government carefully chose the adjective “extreme” to be used for $G > 1$. We have elected to leave “extreme aerodynamics” in the manuscript to describe these violent flows.

4. Introduction Para2: It would be good to acknowledge that UAVs already operate to deliver parcels for example in Australia by SwoopAero, and Google X since 2019. Therefore statements like this needs to be updated since the technology has already started operation, “These new and amazing flying vehicle concepts will likely revolutionize air-based transportation 5, 10, 11”.

Response: Thank you. We have updated our statement and provided citations for Google X and SwoopAero (the second paragraph in the introduction).

5. Introduction Para3: Suggest to not list gusts and vortical disturbances as two separate things. It may be good to reference the gust taxonomy and use the terminology from it:

- Mohamed, A.; Marino, M.; Watkins, S.; Jaworski, J.; Jones, A. Gusts Encountered by Flying Vehicles in Proximity to Buildings. *Drones* 2023, 7, 22. <https://doi.org/10.3390/drones7010022>

Response: Thank you for suggesting this taxonomy. From a Helmholtz decomposition point of view, we believe it is important to distinguish the two but we do see a lot of importance in the taxonomy that is mentioned above. We now cite the suggested paper to highlight the variety of disturbances that can be encountered in gusty conditions.

6. Also the statement of “Such a flight environment has been off-limits due to the fact that there is virtually no available theory for extreme aerodynamic problems and to avoid possible loss of aircraft”, is not accurate since drones have operated in these environments (see references below).

- Mohamed, A., Abdulrahim, M., Watkins, S., & Clothier, R. (2016). Development and flight testing of a turbulence mitigation system for micro air vehicles. *Journal of Field Robotics*, 33(5), 639-660.
- Prudden, S., Fisher, A., Marino, M., Mohamed, A., Watkins, S., & Wild, G. (2018). Measuring wind with small unmanned aircraft systems. *Journal of Wind Engineering and Industrial Aerodynamics*, 176, 197-210.
- Gavrilovic, N., Mohamed, A., Marino, M., Watkins, S., Moschetta, J. M., & Bénard, E. (2018). Avian-inspired energy-harvesting from atmospheric phenomena for small UAVs. *Bioinspiration & biomimetics*, 14(1), 016006.

Response: Thank you for sharing the above papers but as far as we can check and extract from them, the gust ratios considered therein are limited to $G \lesssim 1$, which we do not refer to as extreme aerodynamic

conditions. While we may not have been able to extract some of the values exactly from the figures in those papers, it appears that the strengths of the disturbances are not close to the range that is considered in the present study. Hence, we have not changed our aforementioned statement. Please also note that we are noting the lack of ‘theory’ to call attention to the need for a novel approach. The present study should be considered as a companion to the experimental and computational studies in unsteady aerodynamics. We feel that it is important to cite the suggested references and have now included them in the introduction.

7. Also you don’t need to understand the theory for a controller to mitigate the disturbances which has already been demonstrated for fixed and rotary wings... As mentioned before researchers are even exploring the use of ML to create controllers that use an array of sensors during stall (see below)... The narrative needs to therefore be scoped down and more focused.

- Araujo-Estrada, S. A., & Windsor, S. P. (2021). Aerodynamic state and loads estimation using bio-inspired distributed sensing. *Journal of Aircraft*, 58(4), 704-716.

Response: We find this comment disconcerting as understanding aerodynamics is critical for stable operation of aircraft. We believe having a solid foundation on the aerodynamic response for extreme conditions is very important for future operations of aircraft in such environments. Please note that the major finding here is the discovery of the low-dimensional manifold that captures the complex vortex-airfoil interactions at extreme levels. From the perspective of unsteady aerodynamics, we have tried to ensure that our technical statements in the paper are on the conservative side with accurate descriptions. While we realize that controllers perform well for relatively low level of disturbances, we understand that traditional controllers do not work well for the very high level of disturbances considered in this study. Based on private communications with international stakeholders in the fields of aerodynamics, controls, and sensor and actuators, we strongly believe that developing novel aerodynamic models that can capture the aerodynamic response for extreme level of disturbances is critically needed. We also mention that predictions of aerodynamic forces from sensor measurements are relatively easy, which we have also examined (Zhong et al., 2023; Chen et al., in review). In contrast, estimating the full flow field is exceptionally harder, which is the focus of this work. Please note this paper is focused on the fluid dynamics and not on the control side that the referee is hinting. Having said that, we have upcoming work on our control efforts for this type of flows, which should appear soon.

- Zhong, Y., Fukami, K., An, B., & Taira, K. (2023). Sparse sensor reconstruction of vortex-impinged airfoil wake with machine learning. *Theoretical and Computational Fluid Dynamics*, 37, 269-287.
- Chen, D., Kaiser, F., Hu, J.-C., Rival, D. E., Fukami, K., & Taira K., (2023). Estimating aerodynamic loads in gusty environments: A machine learning approach with sparse pressure data. in review.

8. Introduction Para6 / last sentence: Based on the fact that the focus of the paper is on use of ML to only model a vortex hitting a canonical airfoil as stated in this sentence; its best to rescope the abstract and introduction accordingly since you only considered 1 orientation of flight through a vortex in a 2D sense. There are other flow disturbances (such as shear layers) and even flying through them at different orientations/angles which is important from a 3D perspective that you have not yet explored. There is also the fact that experiencing a single discrete vortex can be different to experiencing a series of vortices of different scales (more realistic) which is probably worth mentioning as a condition to consider in the future.

Response: Thank you for the comment. As demonstrated in figures 5, 6, and 7, we have applied our model trained with only a single-vortex gust to unseen cases that experience multiple vortices. We believe that the present results have demonstrated that the present lift-augmented autoencoder can handle a more challenging and realistic extreme flight condition compared to scenarios with a single vortex gust. The extension to three-dimensional flow fields is definitely one of our future studies. We have added discussions on explorations for three-dimensional flows on page 10.

9. Figure 1 (Right) a scale is missing.

Response: This generic vorticity field is only an illustration of the problem setup. Hence, we have not provided a scale for vorticity. For spatial scales, the reference scale is the chord length, which we noted in the text.

10. Extreme Vortex Airfoil interaction Para1: Can you add any validation of the simulations undertaken to increase confidence in the simulations. I couldn't find it in the Supp material too. Comparing the simulated flow behavior to published experiments is important here.

Response: The simulated flows were validated with previous studies [44,50,51,52], in particular with a study that considered a vortex-airfoil interaction problem [44]. The above statement is added in Section "Simulations of Extreme Vortex-Airfoil Interactions" in the supplemental material (page 1).

- Kurtulus, D. F. (2015). On the unsteady behavior of the flow around NACA 0012 airfoil with steady external conditions at $Re = 1000$. *International journal of micro air vehicles*, 7(3), 301-326.
- Liu, Y., Li, K., Zhang, J., Wang, H., & Liu, L. (2012). Numerical bifurcation analysis of static stall of airfoil and dynamic stall under unsteady perturbation. *Communications in Nonlinear Science and Numerical Simulation*, 17(8), 3427-3434.
- Di Ilio, G., Chiappini, D., Ubertini, S., Bella, G., & Succi, S. (2018). Fluid flow around NACA 0012 airfoil at low-Reynolds numbers with hybrid lattice Boltzmann method. *Computers & Fluids*, 166, 200-208.
- Zhong, Y., Fukami, K., An, B., & Taira, K. (2023). Sparse sensor reconstruction of vortex-impinged airfoil wake with machine learning. *Theoretical and Computational Fluid Dynamics*, 37, 269-287.

11. Figure2: What is unit of Lift on y -axis? Can it be non-dimensionalised? Also I'm not sure what all the light colored curves are on the plot? Finally I assume that the contour represents pressure? Its stated in figure.

Response: Lift here is non-dimensionalized already as stated in Supplemental Material. Due to the editorial (space) constraint with Nature Communications, details of the setup are provided in Supplemental Material, which is standard practice. Please note that the non-dimensional lift coefficient C_L is referred to as 'lift' in the main text, as described in Supplemental Material.

The light-colored curves correspond to all lift responses considered in the present study. The contour plots visualize the vorticity field as described in the caption. To clarify, we have updated the caption of figure 2.

12. Extreme Vortex Airfoil interaction Para3: It would be good to explicitly state the RE here. Also worth noting that atmospheric flow is rarely laminar which affects flow behavior significantly. Acknowledging this is important as part of stating the underlying assumptions and explaining it's a step towards more complex and realistic simulations.

Response: The Reynolds number, Re , was mentioned in paragraph 3. In the revised text, we elaborate on the underlying assumptions for the Reynolds number setting on page 3. For flows with extreme level of unsteadiness, acceleration becomes an important part of the vortex dynamics in terms of injecting a large amount of vorticity and providing a laminar core for vortices, which is the main feature of the problem. Hence, laminar flow models capture the essential physics of vortex-airfoil interactions over a range of Reynolds numbers. We have actually shown this to be true even for three-dimensional flows over a finite wing at a much higher Reynolds number through a major collaboration with experimentalists (Neal et al., 2023).

- Neal, J., Burtsev, A., Ribeiro, J. H. M., Theofilis, V., Taira, K., & Amitay, M. (2023) Effect of Reynolds number on separation over swept and tapered finite span wings. to be submitted.

13. The descriptions of the reconstructed lift isn't very clear. Does this mean that the state variables are used to produce the lift value given the latent state values? If so, why isn't lift just another part of the auto-encoder input/output? Why is the augmentation required?

Response: The lift coefficient C_L is not given as a latent variable. In the present formulation, the nonlinear autoencoder additionally estimates C_L from the latent variables ξ to promote manifold identification such that $[\hat{q}(t), \hat{C}_L(t)] = \mathcal{F}(\mathbf{q}(t))$, as described in the "Autoencoder setup" in Supplemental Material. Based on the lift decoder \mathcal{F}_L and the encoder \mathcal{F}_e , the reconstructed lift coefficient $\hat{C}_L(t)$ is given as

$$\hat{C}_L(t) = \mathcal{F}_L(\xi(t)) = \mathcal{F}_L(\mathcal{F}_e(\mathbf{q}(t))).$$

This expression is now added to the end of page 2 of Supplemental Material.

14. "autoencoder to compress the vorticity field to mere three variables is not only surprising but also reaffirms that the flow field is indeed comprised of common flow features" → not really surprising given autoencoder literature - the whole point is to use few features! the PCA analysis' highlight few field being used.

Response: From a fluid dynamics point of view, this is very surprising. The authors have been performing research in linear and nonlinear compression of fluid flows based on modal analysis and autoencoders for quite some time. We are not aware of other techniques that have been able to achieve significant compressions of the flow field to mere three latent variables (also demonstrated for experimental data at higher Re in another paper of ours in review; Smith et al., 2023), while extracting the low-dimensional manifold for the considered type of problem. We have been communicating this point with other international experts in fluid dynamics and aeronautics and they are also equally surprised. While the referee hints at PCA being able to highlight some features, PCA completely fails to capture the flow field in a low-dimensional manner.

- Smith, L., Fukami, K., Sedky, G., Jones, A., & Taira, K. (2023). A cyclic perspective on transient gust encounters through the lens of persistent homology. arXiv:2306.15829.

15. Images start switching "input" and "reference" image in the figures, but doesn't clearly describe this distinction in the text. Is the reference input? or is this what should be being produced by the network? if so, where does this reference image come from?

Response: In figure 6, the reason why we use "input" for noisy data is, this is post-processed "input data" by adding noise to "reference." "Reference" is used for cases when simulation data are directly fed into an autoencoder. These are standard usages in the data science/applied mathematics literature.

16. Is this just showing the input and output can be replicated? Why is this significant for nature? How this actually applied?

Response: Since the findings in the present study (especially for extrapolating situations shown in figures 5-7) suggest that the discovered manifold is universal for a variety of extreme aerodynamic flow scenarios, it is not a simple replication between a given input and output data. In fact, the present model is able to produce accurate lift responses for unseen cases. This is true even for cases with multiple vortices impacting the wing, despite that the training was performed only with single vortex cases.

These findings provide a novel and powerful foundation for modeling and controlling extreme aerodynamic flows under severe conditions. We already started on these efforts and are achieving promising control results. We have elaborated on these items now in the last section of the paper.

Response to Reviewer 4

Journal: Nature Communications (Manuscript ID: NCOMMS-23-21038)

Title: Grasping Extreme Aerodynamics on a Low-Dimensional Manifold

Authors: Kai Fukami, Kunihiko Taira

Date sent: August 12, 2023

The authors would like to thank the referees for taking his/her time to review our manuscript. Based on the reviewers' insightful comments, revisions have been made to the manuscript, which are highlighted in red. We hope our responses fully address the comments from the referee. In what follows, we list the referee's comments with our responses shown in blue.

.....

This paper reveals that extreme aerodynamic fields can be beautifully embedded in low-dimensional space by data-driven methods. The idea of introducing a lift-augmented structure to the traditional autoencoder is simple and clear: the features are left with enough information to estimate the lift. However, the difference emerged by this modification is significant, and the feature space that the autoencoder composes becomes completely different. In particular, it is surprising that it can be generalized to extrapolative situations such as those shown in the lower part of Figure 5, Figure 6, and Figure 7. The description of the paper is basically clear and well-organized.

Response: Thank you for the encouraging remarks. We have revised our manuscript based on your comments and hope that the updated manuscript addresses your comments.

1. It would be desirable to write more clearly what is to be done with the obtained feature space and why we are happy to have this embedding throughout the text. Several interesting expected specific applications are given in the discussion section, e.g:

1. "low-dimensional representation of the extreme aerodynamic flows suggests that only a small number of sensors on the airfoil may be able to accurately reconstruct the surrounding flow field in real time",
2. "it is possible to develop a reduced-order model that can capture the dynamics in the latent space to desired level of accuracy and complexity",
3. "With the discovered manifold, active flow control and vehicle stability control strategies can be developed for mitigating the effects of extreme aerodynamic disturbances".

It would be helpful for readers to additionally explain the connection between the applications and the proposed method. It would be even better if this explanation could be related to existing works including the authors' ones dealing with flow field data based on autoencoders.

Response: Thank you for the valuable suggestions. In the revised paper, we have now highlighted how the discovered manifold can be leveraged not only for understanding extreme aerodynamic flows but also for aiding downstream tasks such as real-time flow estimation, dynamical modeling, flow control, and vehicle design, throughout the manuscript. In particular, we now have additional discussions in the last section of the paper to present possible future directions based on the present findings, following the reviewer's suggestion. We already started a few efforts on the aforementioned items and have achieved promising results. Please stay tuned: these ongoing works should appear soon.

2. While the title and other text uses word "manifold", can the destination of this embedding be considered as a manifold? It seems to me that it is just a "space" of three dimensions. Are you saying that 3-dimensional ξ space is a manifold in some sense? Or are you referring to the cone-shaped structure? If so, is the structure really within 2 dimensions (even approximately)? From the right side of Fig. 4, it seems to be widely distributed in the three-dimensional space. The claim in the discussion section that the embedding can be

reduced to two variables is also questionable.

Response: Thank you. These are wonderful questions. We have now carefully defined the term “manifold” and used it to refer to an inertial manifold to which the long-time dynamics converge (De Jesús and Graham, 2023). Identified by the lift-augmented autoencoder, the asymptotic periodic shedding states of the airfoil wakes provide the backbone of the inertial manifold with the extreme aerodynamic trajectories lying in the vicinity of this manifold in the three-dimensional latent space. As such, the cone-shaped structure is referred to as the manifold in our manuscript. With the manifold surface being uncovered, the data can be projected on its surface to further reduce the latent space representation to two dimensions from three dimensions. To clarify these points, we describe the above points on page 7.

- De Jesús, C. E. P., & Graham, M. D. (2023). Data-driven low-dimensional dynamic model of Kolmogorov flow. *Physical Review Fluids*, 8(4), 044402.

3. In the abstract, it is stated that “there is an enormous parameter space for gusty conditions wings encounter”. What does this mean? I could not find a detailed description in the main text. Do you mean the space of four parameters (or, plus Re) in the experiment? Or do you mean a much larger space of parameters? If the latter, is 4 or 5 dimensions in the authors’ experiment sufficient?

Response: Thank you for requesting this clarification. What we intended to convey for the description of the parameter space is the very large combinations of parameters that would be needed to capture the full dynamics of extreme level of disturbance impacting a wing. These parameters are comprised of G , α , L , and y_0/c in the current study (in reality, the number of parameters increase significantly more with Re , orientation, wing geometry, noise level, multiple vortex arrangement, etc.). Strictly speaking, there would be an infinite number of combinations needed to continuously describe the physics over the parameter space. This is what we described as an enormous parameter space. We have rephrased “enormous parameter space” to “large parameter space” to avoid readers from receiving a different impression. These points are mentioned on page 3 of the revised text.

4. The abstract states that “the fundamental physics behind extreme aerodynamics is far simpler and low-rank than traditionally expected”. How much dimensionality has conventionally expected by whom? It should be stated in the main text.

Response: Thank you for requesting the clarification. In the flow modeling community, it is well-known that a linear projection-based reduced-order model for an (steadily) oscillatory laminar bluff-body flow requires at minimum about 4 spatial modes (e.g., POD/PCA) in addition to the mean flow. Incorporating a sliding scale to develop a composite reduced-order model from different angles of attack, it would in total require at minimum 26 modes. Such a composite model unfortunately would not even capture disturbances interacting with the airfoil, thus not performing anywhere close to the level of the present autoencoder-based technique. We have added a short description of these points on page 5 now.

5. The Reynolds number used in this study is fixed at 100. What specific application is this Reynolds number set with in mind? Is it not too low? The following sentences in page 3: “The current choice of Re captures flows that are representative of a wide range of realistic flow scenarios since the extreme aerodynamic disturbance creates high local flow acceleration injecting a large amount of vorticity from the surface. This generally leads to laminarizing the flow near the body.” may be in response to this question, but if so, could you provide more detailed explanation?

Response: Thank you for the comment. The choice of Reynolds number is not too critical for the current two-dimensional problem setup since the flow is dominated by strong local acceleration that generates vorticity from the body surface. The setup also enables us to capture the dynamics of the large vortex core (which tends to be laminar even at much higher Re) interacting with the wing. Hence, we believe the choice of $Re = 100$ is appropriate for this paper. Please see our additional explanation on page 3.

6. In page 5, the authors stated that “While powerful at compressing Gaussian-distributed data, PCA is known to have difficulty in compressing transient physics and those with strong nonlinearities that produce non-Gaussian distributions, which is certainly the case with extreme aerodynamic flows”, which seems unclear to me. It is true that PCA can be regarded as a probabilistic modeling with multidimensional normal distribution, but, in my understanding, Gaussianity and the compression performance is almost unrelated. What determines the compression rate by PCA is how much data lies in a limited linear subspace. The reason AE works is that this subspace can be bent by nonlinear transformations. In addition, I don’t understand well the term “nonlinearity”. The meaning of the term nonlinear seems unclear throughout the main text. What is nonlinear object in this sentence, for example? Do you mean that the Navier-Stokes equation is a nonlinear differential equation? Is it OK to discuss the linearity/nonlinearity of PCA/AE transformations and the nonlinearity of differential equations as the same thing?

Response: Thank you. We should have been more careful with the description as pointed out by the referee. In the revised text related to the discussions on POD/PCA, we have removed some of the comments that were problematic.

7. Figures 5, 6 and 7 show the structural similarity index. Since it is not a general index value, it should be mentioned briefly in the text.

Response: We have added an expression for the structural similarity index in the main text. Thank you for the suggestion.

8. In page 7, the authors stated that “it can also be considered as an hour-glass shape since there is a mirrored manifold for negative angle of attack cases.” Since AE performs nonlinear transformations, I don’t think that simply mirroring the geometry works in general. Have you tried embedding the negative angle of attack cases? If it works, it is rather interesting result and should be further discussed.

Response: This is a good point. We agree that the distributions of the trajectories are not guaranteed to be a simple mirroring due to nonlinear transformation. We however have confirmation from a companion study that reveals the hour-glass like manifold geometry for negative angles of attack. We have not shown this result since the companion study is slightly out-of-scope.

REVIEWERS' COMMENTS

Reviewer #1 (Remarks to the Author):

I appreciate authors' efforts to revise the manuscript.

Most of my concerns are addressed. I checked the Discussion section and I think this revision makes this paper better.

A minor point related to the latent dimension number. I recommend including figures or citations that demonstrate latent dimension = 3 is a good choice. For example, the authors may show when setting latent dimension = 2 is not reconstructing well, and when latent dimension = 4, the performance is around the same. I understand that conducting additional experiments may be challenging. But this could offer valuable insights for the community.

Reviewer #2 (Remarks to the Author):

I don't have any further comments, responses to my comments are satisfactory.

Reviewer #4 (Remarks to the Author):

Thank you for your thoughtful response to the comments. Your answers helped me a lot in my understanding.

Since responses and corrections have been made satisfactory to my comments, I believe the paper deserves to be published as is.

Response to Reviewer 1

Journal: Nature Communications (Manuscript ID: NCOMMS-23-21038A)

Title: Grasping Extreme Aerodynamics on a Low-Dimensional Manifold

Authors: Kai Fukami, Kunihiko Taira

Date sent: September 15, 2023

The authors would like to thank the referees for taking his/her time to review our manuscript. Based on the reviewers' insightful comments, revisions have been made to the manuscript, which are highlighted in red. We hope our responses fully address the comments from the referee. In what follows, we list the referee's comments with our responses shown in blue.

.....

I appreciate authors' efforts to revise the manuscript. Most of my concerns are addressed. I checked the Discussion section and I think this revision makes this paper better.

A minor point related to the latent dimension number. I recommend including figures or citations that demonstrate latent dimension = 3 is a good choice. For example, the authors may show when setting latent dimension = 2 is not reconstructing well, and when latent dimension = 4, the performance is around the same. I understand that conducting additional experiments may be challenging. But this could offer valuable insights for the community.

Response: Thank you for the encouraging remarks. We have included a citation (Smith et al.) that examines the influence of the latent space size.